# Recent Advances in Microbiota-Associated Metabolites in Heart Failure

**DOI:** 10.3390/biomedicines11082313

**Published:** 2023-08-21

**Authors:** Sepiso K. Masenga, Joreen P. Povia, Propheria C. Lwiindi, Annet Kirabo

**Affiliations:** 1HAND Research Group, School of Medicine and Health Sciences, Mulungushi University, Livingstone Campus, Livingstone 10101, Zambia; joreenpovia@gmail.com (J.P.P.); pclwiindi@gmail.com (P.C.L.); 2Department of Medicine, Vanderbilt University Medical Center, Nashville, TN 37232-6602, USA

**Keywords:** heart failure, microbiota, gut metabolites, hypertension, intestinal flora, bile acids, short chain fatty acids, branched chain amino acids, tryptophan, indole derivatives

## Abstract

Heart failure is a risk factor for adverse events such as sudden cardiac arrest, liver and kidney failure and death. The gut microbiota and its metabolites are directly linked to the pathogenesis of heart failure. As emerging studies have increased in the literature on the role of specific gut microbiota metabolites in heart failure development, this review highlights and summarizes the current evidence and underlying mechanisms associated with the pathogenesis of heart failure. We found that gut microbiota-derived metabolites such as short chain fatty acids, bile acids, branched-chain amino acids, tryptophan and indole derivatives as well as trimethylamine-derived metabolite, trimethylamine N-oxide, play critical roles in promoting heart failure through various mechanisms. Mainly, they modulate complex signaling pathways such as nuclear factor kappa-light-chain-enhancer of activated B cells, Bcl-2 interacting protein 3, NLR Family Pyrin Domain Containing inflammasome, and Protein kinase RNA-like endoplasmic reticulum kinase. We have also highlighted the beneficial role of other gut metabolites in heart failure and other cardiovascular and metabolic diseases.

## 1. Introduction

Heart failure is a clinical syndrome characterized by typical signs and symptoms caused by a structural and/or functional cardiac abnormality leading to a reduction in cardiac output and/or elevated intra cardiac pressure at rest or during stress [1]. some clinical features of heart failure include fatigue, peripheral edema, elevated jugular venous pressure and shortness of breath [1,2]. It is estimated that heart failure affects up to 64 million people worldwide [3]. There are a number of risk factors associated with heart failure. Some common ones include sedentary lifestyle, hypertension, diabetes, smoking and hyperlipidemia [4].

The gut microbiota is a dynamic integral part of the human body acquired at birth that performs some basic functions in its metabolic, structural, neurological and immunological landscape as well as exerting significant influence on physical and mental health [5]. Studies have shown that the gut microbiota can either directly or indirectly be involved in the pathogenesis and progression of cardiovascular diseases (CVDs), heart failure inclusive [6,7]. Both the gut microbiota and its associated metabolites have been implicated in heart failure [6,8,9]. CVDs such as hypertensive heart disease, atherosclerosis, myocardial infarction, heart failure and arrhythmia have been associated with altered intestinal flora [10,11]. Furthermore, gut microbial fermentation metabolites have been implicated in the development, prevention, treatment and prognosis of CVDs and these include trimethylamine N-oxide (TMAO), short chain fatty acid (SCFA), secondary bile acid (BA) and gases such as hydrogen sulfide (H₂S), carbon dioxide (CO_2_) and nitric oxide (NO) [10,12]. The association between gut microbiota and the biological processes affecting CVD risk is complex [13]. However, there is paucity of data on the effects of gut microbiota-associated metabolites in heart failure and hence the need for an in-depth review to understand their role in heart failure to help develop and accelerate therapeutic potential in the future. In this review, we discuss the current evidence on the role of gut microbial metabolites in the pathogenesis of heart failure. We also discuss the potential benefits and dietary interventions available to modulate the gut microbiota in order to promote cardiovascular health.

## 2. Heart Failure Global Burden and Quality of Life

More than 64.3 million people had heart failure globally by 2017 accounting for an estimated age-standardized prevalence of 831.0 and 128.2 per 100,000 persons of years lived with disability (YLDs) [14]. The highest prevalence rates were reported from Central Europe, North Africa, and the Middle East [14,15]. By 2019, the prevalence rates of heart failure were estimated to be 17 per 1000 persons across 13 European countries [16].

In the USA, based on the National Health and Nutrition Examination Survey (NHANES) data, about 6.0 million adults aged 20 years and above had heart failure accounting for a prevalence of 2.4% of Americans living with heart failure by 2012 [17]. The costs associated with the management of heart failure has been rising over the years and in the USA alone, about $30.7 billion was used to manage heart failure patients in 2012 with a suggested 127% increase to $69.8 billion by 2030 [17,18]. Generally, the economic global burden of heart failure is substantial and has been rising over the years. In 2012 alone, an estimated total cost of more than $108 billion was spent worldwide [19].

The quality of life for patients with heart failure varies globally. Health-related quality of life (HRQL) of patients with heart failure was lowest in Africa and highest in Western Europe (mean ± SE, 39.5 ± 0.3 vs. 62.5 ± 0.4, respectively) with 4460 (19%) deaths and 3885 (17%) hospitalizations due to heart failure from 40 countries in eight different world regions [20]. Impairments in physical function and cognition, depression and reduced quality of life (QoL) is severely marked in older acute decompensated heart failure patients above 60 years of age [21]. In the USA, a study evaluating 15 chronic conditions reported that participants with heart failure had the lowest age-adjusted quality-adjusted life years (QALYs) with significant differences in sex whereby men had lower age-adjusted QALYs compared to women (1.1–1.5 vs. 1.5–2.2 years, respectively) [22]. Data from the 1998–2014 waves of the Health and Retirement Study showed that the weighted overall disability-adjusted life years (DALYs) for heart failure and hypertension was 62,630 and 378,849 years, respectively, for middle-aged and older adults in the United States [23]. With inadequate measures to control and prevent hypertension, gains to productivity are adversely affected as workers are either absent from work or suffer from presenteeism (reduced efficiency at work) [24]. This reduces Productivity-Adjusted life years (PALY) and causes loss to Gross Domestic Product (GDP) [25].

Overall, these data reveal the immense burden and importance of heart failure globally. Therefore, efforts to understand the risk factors, underlying mechanisms and management remains of paramount importance.

## 3. Gut Microbiota Species Implicated in Heart Failure

Gut microbiota belonging to any of the three domains of life that include *Archaea, Eukarya* and *Bacteria* immediately colonize the gut of a newborn after birth to establish a relationship with each other and their host, and this relationship may be symbiotic, or parasitic if the gut ecosystem and microbiota homeostasis is disturbed [26]. Mostly, three phyla of bacterial species are found in the human gut and these include *Bacteroidetes* (*porphyromonas*, *prevotella*), *Firmicutes* (*Ruminococcus*, *Clostridium* and *Eubacteria*) and *Actinobacteria (Bifidobacterium)*. *Lactobacilli, Streptococci* and *Escherichia coli* are also found in small numbers [27]. However, there is a decrease in specific gut microbiota such as *Bifidobacteria* observed in heart failure patients. Other species implicated in heart failure include but are not limited to *Salmonella*, *Shigella*, *Escherichia*, *Campylobacter*, *Klebsiella*, *Yersinia*, *Candida* and *Clostridium difficile* [28,29,30]. The gut microbiota may either have a symbiotic or dysbiotic interaction with their host [31]. Dysbiosis, a disturbance or imbalance in the gut microbiota components, has been associated with pathological conditions. However, in a symbiotic relationship, microbiota benefit from the nutrient rich environment that the gut provides and in turn the microbiota produce metabolites associated with key host functions such as immune development, maintenance of homeostasis and nutrient processing [31]. A reduction in microbial diversity accompanied by immune cell activation leads to an imbalance of microbial species within the gut and this has been implicated in the pathogenesis of heart failure [32]. The microbiota change in composition of species has been well summarized elsewhere [33]. Hence, we will focus more on elucidating the mechanisms through which gut metabolites contribute to the pathogenesis of heart failure.

## 4. Mechanisms of Gut Microbiota Metabolites Implicated in Heart Failure

Many recent studies have shown the importance of the gut microbiota and its metabolites in the pathophysiology of heart failure. The gut microbiota has been reported to play a role in increasing gut permeability to facilitate the translocation of microorganisms into the bloodstream, ultimately leading to low-grade chronic inflammation [6]. Like an endocrine organ, the gut microbiota generates bioactive metabolites that cause changes to the physiology of the host through interactions via different pathways including the short chain fatty acid (SCFA) pathway, bile acid (BA) pathway, Trimethylamine N-oxide (TMAO)/Trimethylamine (TMA) pathway, etc., and their presence in circulation increases the inflammatory milieu thereby contributing to the progression of heart failure [34]. Although not quite clear, the relationship between the gut microbiota and heart failure is bidirectional and may likely involve “a leaky gut–heart axis theory” where alterations in the structure and function of the gut lining from other causes facilitates the translocation of bacteria into the systemic circulation resulting in systemic inflammation, and whereby this bacteria translocation to the heart elicits structural and functional changes to heart tissue contributing in this way to heart failure [35]. On the other hand, reduced perfusion of tissue including microcirculatory insufficiency to perfuse the gut due to heart failure may alter the structure and function of the gut lining leading to disruptions in intestinal epithelial gap junctions and microbial translocation, thus creating a vicious cycle [35]. Another important underlying mechanism that promotes the pathogenesis of heart failure is local and systemic inflammation [36]. Several inflammatory mediators are found to be elevated in heart failure. These include transforming growth factor-β (TGF-β), soluble interleukin (IL)-1 receptor-like 1, tumor necrosis factor alpha, soluble tumor necrosis factor receptor type I (sTNFRI), growth differentiation factor 15, IL-6, soluble ST2, pentraxin-3, et cetera [37,38,39]. Inflammatory mediators promote endothelial dysfunction, myocardial fibrosis, arterial stiffness via direct activation of fibroblasts and recruitment of activated macrophages that promote myocardial cell necrosis and fibrosis [40,41,42,43]. One of the underlying intracellular mediators of inflammation that is activated in heart failure is the NLR family pyrin domain containing 3 (NLRP3) inflammasome. The NLRP3 inflammasome promotes disruption of tight junctions in endothelial cells resulting in endothelial dysfunction and may also induce ventricular arrhythmias in heart failure with preserved ejection fraction [44,45]. It is interesting that inhibiting the NLRP3 inflammasome reduces inflammation, hypertrophy, fibrosis and reverses pressure overload-induced pathological cardiac remodeling [46]. Another underlying mechanism that promote the pathogenesis of heart failure is the activation of the calcineurin–nuclear factor of activated T cells (NFAT) signaling pathway which initiates the transcription of multiple genes responsible for promoting cardiac hypertrophy; however, inhibition of calcineurin by cyclosporin A, although controversial, is effective in reversing these effects [47,48]. Other mechanisms involved in the pathophysiology of heart failure, some of which will be discussed in detail later, include G protein-coupled receptor-mediated signaling [49], Mitogen-activated protein kinase (MAPK) signaling, phosphoinositide 3-kinases (PI3K)-AKT signaling, Wnt signaling, and pathways associated with cardiomyocyte death such as apoptosis, necroptosis, pyroptosis, autosis, and ferroptosis [50].

### 4.1. Beneficial Effects of Gut Microbiota-Derived Metabolites in Heart Failure Pathophysiology

Many studies available have elucidated the harmful effects of the gut microbiota metabolites in the pathogenesis of heart failure with little emphasis on their benefits [35,51,52,53,54]. In addition to the few metabolites that we shall discuss, it is important to understand that several gut metabolites are beneficial in the pathophysiology of heart failure. For example, polyphenols, which are derived from plant-based diets, are metabolized into bioactive compounds/metabolites by the gut microbiota to promote their beneficial function on cardiovascular health [55]. Some of the phenolic compounds’ beneficial effects are linked to their properties and these include antioxidant, anti-inflammatory, antibacterial [56,57], anti-adipogenic [58,59,60] and neuro-protective properties [55,61,62]. The hydroxyl group situated on the benzene ring of polyphenols functions to mediate the transfer of the H-atom to free radicals converting them into non-toxic compounds and thus effecting their antioxidant function [63]. In terms of anti-inflammatory function, polyphenols are able to suppress pro-inflammatory cytokines such as IL-1β, IL-6, TNF-α, IFN-γ, IL-1α, and IL-4 through mechanisms that are still not clear but related to inhibiting NF-κB [64,65,66,67]. The overall resulting effect of polyphenols on the heart is suppression of fibrotic and hypertrophic processes, reduction of free radical production and regulation of cellular metabolism to prevent heart failure [68]. More on this is discussed in later sections below.

### 4.2. Gut Metabolites Implicated in Heart Failure

The gut microbiota plays a role in further digestion of carbohydrates, proteins and, to a lesser extent, fat and other biomolecules including fermentation of non-digestible substrates [69]. Some of the metabolites produced in this process have been implicated in heart failure. The most important are discussed below in detail.

### 4.3. Short Chain Fatty Acids (SCFAs)

Fermentation of resistant starch and dietary fiber such as pectin, cellulose and lignin by the gut microbiota produces saturated fatty acids known as short chain fatty acids (SCFAs) made up of six or less carbon molecules, and these include valeric acid, caproic acid, acrylic acid, acetic acid, propionate, and butyric acid [33]. Acetic acid or acetate, propionate, and butyrate are the most common SCFAs resulting from microbiota metabolism [12]. SCFAs provide energy for intestinal epithelial cells and are also involved in metabolic, gut barrier integrity, appetite, gut hormone production, stimulation of water and sodium absorption, and immune and inflammatory responses as signaling molecules [12,70]. SCFAs stimulate host G-protein coupled receptor 41 and 43 (GPR 41, GPR 43) pathways that impact on renin secretion and the regulation of blood pressure [71]. Although the expression of GPR 41 and GPR 43 in the heart is low, signaling through these GPRs is known to be protective against heart failure [72,73]. For example, signaling through the endothelial GPR41 has been reported to lower blood pressure by decreasing vascular tone in blood vessels [73]. In both animal and human studies SCFAs have been reported to have blood pressure lowering effects through their vasodilatory effects, and supplementation with acetate, butyrate, and propionate prevented an increase in blood pressure [72,74,75,76,77].

SCFAs such as butyrate and propionate have been known to be key regulators of pro-inflammatory innate immune responses by inhibiting histone deacetylases (HDAC) [78]. Further, several studies have shown that exposure of neutrophils and other peripheral blood mononuclear cells to SCFAs suppresses nuclear factor kappa-light-chain-enhancer of activated B cells (NF-κB) and decreases the generation of the pro-inflammatory cytokine tumor necrosis factor alpha (TNF-α) which is consistent with their response to other HDAC inhibitors [79]. NF-κB is a well-known player in mediating inflammation and cardiac and vascular damage and is activated in cardiomyocytes and innate cells in many heart conditions including heart failure [80,81,82,83]. For example, NF-κB was significantly activated in peripheral leukocytes of patients with stable heart failure [84] suggesting that it plays a potential significant role. However, NF-κB has also been reported to play a cardioprotective role in acute hypoxia by synergizing with HDAC to inhibit the hypoxia-inducible death factor Bcl-2 interacting protein 3 (BNIP3) in ventricular myocytes [85]. However, prolonged activation is detrimental to the failing heart due to the chronic inflammation that leads to increased production of TNF-α, IL-1, IL-6 whose effects result in endoplasmic reticulum stress responses and cell death [86], Figure 1. TNF increases the permeability of the endothelium and expression of adhesion molecules, thereby promoting the recruitment of leukocytes, upregulating the synthesis of inflammatory and pro-apoptotic cytokines that enhance inducible nitric oxide synthase (iNOS) [36]. iNOS is found in cardiac myocytes, endocardium, vascular smooth muscle cells and infiltrating inflammatory cells. In the presence of cytokines such TNF-α and Interleukins (IL-1, 2 and 6), iNOS is capable of producing large amounts of nitric oxide (NO) which has been implicated as an important negative inotrope hence increasing the progression of heart failure [87].

Generally, SCFAs are beneficial in heart failure as they provide a substantial amount of energy for use by the failing myocardium [88,89]. This is especially true about acetate [88] and butyrate [90]. Butyrate has been reported to be able to reverse and improve mitochondrial ATP production to enhance heart contractility in the failing heart [90]. In this way, SCFAs play a key role in restoring mitochondrial function [91]. Propionate on the other hand regulates energy consumption and enhances sympathetic nervous system via the GPR41 receptor [92]. Although SCFAs are beneficial in heart failure, their potential usage in clinical settings is still under investigation. Their role in heart failure has been extensively discussed elsewhere [78].

### 4.4. Bile Acids

Bile acids are an important part of bile [70]. In humans, the major bile acids include chenodeoxycholic acid, cholic acid (CA), and lithocholic acid and their formation involves at least 17 enzymes [93]. They are synthesized from cholesterol via either the classic or neutral pathway and the alternative or acidic pathway [94]; they are then conjugated in the liver, and secreted into the gut lumen where microbiota metabolizes them into secondary bile acids [95]. Bile acids are involved in metabolism of cholesterol, lipids and glucose as well as absorption of fat [96]. The gut microbiota is maintained in a state of balance under normal physiological conditions and is involved in the formation and regulation of the intestinal mucosal barrier, controlling nutrient intake, storage and metabolism, assisting in immune tissue maturation and preventing the growth of pathogenic microorganisms. However, changes in the bile acid pool can affect gut flora distribution causing pathogenic microorganisms to thrive and lead to pathologic conditions such as inflammatory bowel syndrome, obesity, diabetes, colorectal cancer and CVDs including heart failure [70]. On the other hand, changes in microbiota content can affect the bile acid pool and contribute indirectly and directly to cardiometabolic disease [97]. Bile acids exert inotropic, lusitropic and chronotropic effects when they interact with bile acid receptors such a muscarinic M2 receptor, takeda G-protein-coupled receptor 5 (TGR5) and farnesoid X receptor (FXR) expressed on cardiomyocytes [96]. These receptors for bile acids seem to be activated when secondary bile acids are formed through the presence of specific gut microbiota species [93]. The metabolism of bile acids in relation to cardiometabolic disease has been extensively reviewed elsewhere [97] and we will not discuss it here but will instead highlight the role of secondary bile acids in contributing to heart failure. The role of bile acid–gut microbiota interaction in contributing to heart failure is complex involving multiple pathways, and most studies available in literature were conducted in animals.

Generally, bile acids exert a protective role on heart cells, but some bile acids may exert negative effects. For example, we know that hydrophobic bile acids such as lithocholic acid are toxic to cells and have been implicated in cardiometabolic diseases due to their high affinity for lipids, while hydrophilic bile acids such as ursodeoxycholic acid have beneficial effects on the heart by ameliorating myocardial fibrosis [98,99,100]. Ursodeoxycholic acid binds to FXR to block nitric oxide synthase inhibitors and acts in this way to enhance myofilaments and myocardial relaxation in heart failure with preserved ejection fraction [101]. In mice, binding of bile acids to TGR5 inhibits the NLRP3 inflammasome activation thus preventing inflammation, and also enhances the heart’s ability to adapt to hemodynamic stress in heart failure via activation of pro-survival kinases and heat shock proteins [102,103].

Bile acid receptors FXR and TGR5 play an important role in heart failure. For example, activation of FXR receptors by secondary bile acids in rats can improve the bile acid ratio as well as inhibit the activation of NF-κB to prevent inflammation and hypertrophic changes in the myocardium [104]. Prolonged activation of NF-κB increases atrial natriuretic factor expression and promotes cardiomyocyte enlargement [104]. NF-κB is an important transcription factor that enhances expression of many genes including those involved in inflammation, cell differentiation, proliferation and cell death [105]. In the cytoplasm of quiescent cells, NF-κB dimers are bound to inhibitory proteins (IκB), mainly IκBα and IκBβ [106]. NF-κB is activated when IκB proteins are phosphorylated on specific serine residues by the IκB kinase (IKK), a complex protein composed of an α and β subunit and a regulatory γ subunit, resulting in the degradation of IκB proteins by the 26S proteosome in an ubiquitination-dependent protein kinase activity manner and the release of NF-κB, which then translocates into the nucleus to activate several genes including those involved in the production of pro-inflammatory cytokines [104,107], Figure 2. The bile acid activation of TGR5 in mice has shown to improve cardiac contractility and response to hemodynamic stress [102]. Therefore, when the gut microbiota content is disturbed (gut dysbiosis), that is, when there is reduction of important species that promote good bile quantity and homeostasis including activation of the FXR and TGR5 receptors, this results in increased proinflammatory cytokines, reduced cardiac function and increased oxidative stress in myocardial cells [108]. Thus, modulating the gut microbiota composition has potential to ameliorate and prevent pathological processes that contribute to heart failure [109].

### 4.5. Branched-Chain Amino Acids

Amino acids are the building blocks for protein synthesis [110]. They are key nutrients for the growth, survival and function of cells. Some amino acids include branched-chain amino acids (BCAAs) which have been shown to possess signaling functions that regulate growth and metabolism [111]. BCAAs such as leucine, isoleucine and valine are nutritionally essential amino acids, hence must be acquired from food, and they serve as significant sources for the biosynthesis of sterol, keto bodies and glucose [112]. In addition to these roles, BCAAs also decrease proteolysis, they are used as energy substrates in stress illness, increase glutamine and alanine release from muscles, and reduce adiposity [113]. Several studies have shown that the gut microbiota likely contributes, to a smaller extent, to the synthesis of BCAAs using different nitrogen sources [114]. A few examples of microorganisms that participate in the biosynthesis of BCAAs include *Clostridium species* (spp.), *Staphylococcus aureus*, *Escherichia coli*, *Klebsiella* spp., *Streptococcus* spp., *Selenomonas ruminantium*, *Megasphaera elsdenii*, *Prevotella* spp., and *Bacteroides* spp. [115,116,117]. The amount of BCAA available is to a larger extent determined by dietary composition which in turn determines the type of gut microbiota promoted to metabolize BCAAs. A diet rich in carbohydrates prohibits protein fermentation, while a fat-rich diet promotes BCAA synthesis by stimulating changes in the gut microbiota composition [118,119]. On the other hand, a protein rich diet increases several microbiota species that promote BCAA degradation such as *Eubacterium*, *A. putredinis* spp., *Bacteroides* spp., *Fusobacterium*, *Proteobacteria*, *Bacteroides*, *Proteobacteria*, *Desulfovibrio*, *Bilophila wadsworthia*, *Clostridium* and *Ruminociccu* [120]. In some instances, reduced species in the gut such as *Firmicutes*, *Selenomonas*, *Archaea*, *Megasphera*, *Acidaminococcus*, *Bifidobacterium* and *Prevotella* have been associated with promotion of BCAA synthesis [120].

As oxidation of amino acids is a potential source of ATP production by the heart, BCAAs are the best characterized source. Their metabolism involves transamination of their corresponding branched-chain alpha-keto acids (BCKAs) by the mitochondrial branched-chain amino-transaminase [121]. This is followed by oxidative decarboxylation of BCKA by the mitochondrial branched-chain alpha-keto acid dehydrogenase (BCKDH). The products of BCKDH either generate acetyl-CoA for the TCA cycle or succinyl-CoA for anaplerosis [121]. BCAAs play an important role in regulating biomolecule, nutrient and immune signaling pathways in the heart and many areas of the body, some of which include phosphoinositide 3-kinase/protein kinase B/mammalian target of rapamycin (PI3K/AKT/mTOR) signaling pathways [111]. The mTOR signaling pathway plays a key role in contributing to several cell processes, and dysregulation of the mTOR pathway has been implicated in promoting many diseases including cancer, diabetes mellitus, ageing and cardiovascular diseases [122,123,124]. mTOR is a serine/threonine protein kinase that forms two distinct complexes namely mTOR complex 1 (mTORC1) and mTOR complex 2 (mTORC2) [122]. The two complexes have similar functions with only a few differences. Generally, the mTOR signaling pathway regulates cell processes such as lipid synthesis, autophagy, cell survival, growth and proliferation, mitochondrial function, cell architecture and polarity among others [125,126,127,128]. Although partial inhibition of mTORC1 is cardioprotective in cardiac stress and aging, disruptions in the mTORC1 leads to failure of the myocardium to compensate for hemodynamic pressure load and stress, worsening or leading to heart failure complications [129]. However, partial inhibition may also ameliorate hypertrophic changes and pressure overload in heart failure because of preserved physiological function of mTORC1 while the maladaptive detrimental effects are eliminated during cardiac stress [130,131]. Dysregulation of the gut microbiota that results in dysbiosis has the potential to activate the mTOR pathway promoting derangements in cardiac response to hemodynamic stress and remodeling [132]. Moreover, dietary protein composition does affect the gut microbiota composition, mTOR activity and the transcription of mTOR signaling pathways in the small intestine [133].

Generally, high levels of circulating BCAAs have been associated with cardiovascular disease risk and increased carotid intima-media thickness [134,135]. Increased BCAAs can also promote insulin resistance by either persistent mTOR signaling that results in impaired insulin signal transduction through insulin receptor substrate (IRS) or by increased accumulation of their metabolites resulting in toxic effects [121]. The effect of BCAA on the heart is complex. Accumulation of BCAAs and their metabolites referred to as branched-chain keto acids (BCKAs), resulting from impaired BCAA oxidation, promotes pathological cardiac remodeling mediated by the mTOR signaling pathway [136]. This suggests that interventions aimed at reducing the accumulation of BCAAs and BCKAs would have beneficial effects on the failing heart [136].

Other mechanisms underlying the association between BCAAs and heart failure include mitochondrial dysfunction, cardiac substrate utilization disturbances and inappropriate platelet activation [137]. Accumulating BCAAs and their metabolites is not just detrimental in heart failure but there is a lot of emerging evidence now implicating BCAAs in various metabolic disorders including obesity, insulin resistance, diabetes mellitus, maple syrup urine disease, and hypertension [113,137,138]. However, more studies are required to understand the underlying mechanisms of heart failure associated with BCAAs.

### 4.6. Phenylacetylglutamine

Phenylacetylglutamine is associated with the presence and severity of heart failure [139]. Phenylacetylglutamine is a metabolite of the gut microbiota derived from its nutrient precursor metabolite phenylalanine, a nutritionally essential amino acid that is converted to phenylpyruvate in the gut [140]. Phenylacetylglutamine is catabolized by the gut microbiota to form phenylpyruvate and phenylacetic acid. Phenylacetylglutamine in the liver is then formed from phenylacetic acid and glutamine in an amino acid acetylation process catalyzed by the liver enzyme phenylacetyltranferase or glutamine N-acetyl transferase [141], Figure 3. Phenylacetyltranferase or glutamine N-acetyl transferase catalyzes the reaction of the substrates phenylacetyl-CoA and L-glutamine to produce CoA and alpha-N-phenylacetyl-L-glutamine and phenylacetic acid [142,143].

By interacting with GPCRs and adrenergic receptors (ADRs), phenylacetylglutamine has been shown to impact thrombosis potential by enhancing platelet function that results in hyperresponsive platelets, leading to myocardial infarction in coronary heart disease [140,144]. The interaction of phenylacetylglutamine with GPCRs and ADRs also contributes to the over activity of the sympathetic nervous system, thereby exacerbating heart failure [145]. In a large recent clinical trial, Romano et al. found that circulating plasma levels of phenylacetylglutamine were not only dose-dependently associated with heart failure but also with indices of severity namely reduced ventricular ejection fraction and elevated N-terminal pro-B-type natriuretic peptide [139]. Their study strongly suggests a clinical and mechanistic link between heart failure and the gut microbiota metabolite phenylacetylglutamine. In their study, Romano et al. also showed that phenylacetylglutamine contributes to heart failure by decreasing cardiomyocyte sarcomere contraction and B-type natriuretic peptide gene expression [139]. Another recent study, where they used 16S rRNA sequencing methods to study patients with coronary artery disease (CAD), Fang et al. found that dysbiosis and elevated levels of enhanced microbiota-derived phenylacetylglutamine synthesis was significantly associated with in-stent stenosis and hyperplasia in patients with CAD [146]. Although studies on phenylacetylglutamine are few, emerging evidence that links this gut metabolite to several CVDs highlight the potential therapeutic target of modulating this gut microbiota-derived metabolite to ameliorate CVDs [147]. Especially in heart failure, elevated levels of phenylacetylglutamine are an independent risk biomarker for development of heart failure and consequential adverse events such as renal failure and death [145,148]. Thus, phenylacetylglutamine is a prognostic and risk factor of heart failure. Moreover, in some studies phenylacetylglutamine has also been reported to be a risk factor for acute ischemic stroke [149], white matter hyperintensity in patients with acute ischemic stroke [150], coronary atherosclerotic severity [151] and coronary artery disease [152] and lethal prostate cancer [153].

### 4.7. Tryptophan and Indole Derivatives

Tryptophan is an important monoamine neurotransmitter that plays a role in the regulation of central neurotransmission as well as intestinal physiological function [154]. It is a nutritionally essential amino acid utilized for protein synthesis [155]. The gut microbiota directly metabolizes tryptophan into indole and indole derivatives such as indole-3-acetic acid, indole-3-acetaldehyde, indole-3-aldehyde, indole-3-acetamide, indole-3-lactic acid and indole-3-propionic acid [156]. Other pathways for tryptophan metabolism include the kynurenine pathway (mainly occurring in the liver and to a lesser extent in the brain and gastrointestinal tract) and the serotonin pathway [156].

Indole-derived metabolites are produced through fermentation by *Clostridium sporogenes* and *Escherichia coli* [154]. Indole derivatives act as endogenous ligands of transcription factors that interact with several regulatory and signaling pathways hence mediating cardiotoxicity and vascular inflammation [157].

Tryptophan can be metabolized into indoxyl sulfate which is among the most studied uremic toxins having negative renal remodeling effects and the potential to contribute to cardiac remodeling effects as well, through direct pro-fibrotic, pro-hypertrophic and pro-inflammatory effects [96]. High indoxyl sulfate levels have been shown to worsen diastolic dysfunction as well as cardiovascular events through activation of the renin angiotensin receptor and inducing oxidative stress in endothelial and vascular smooth muscle cells thereby increasing the progression of heart failure [96]. Indoxyl sulfate contributes to heart failure by altering multiple NADPH oxidase-mediated redox signaling pathways that have been linked not only to heart failure but to other CVDs including arrhythmia, atherosclerotic vascular disease and coronary calcification [158].

A recent study has reported that a microbiota-derived tryptophan metabolite, indole-3-propionic acid, is beneficial in heart failure by reducing oxidative stress, cardiomyocyte death and inflammation via inhibition of histone deacetylase 6/NADPH oxidase 2 (HDAC6/NOX2) signaling [159]. The study by Gesper et al. clarifies the role of indole-3-propionic acid in a systematic study where they reviewed several studies and found that indole-3-propionic acid modulated mitochondrial function in cardiomyocytes; however, while acute treatment was beneficial in improving maximal mitochondrial respiration and improved cardiac contractility, long term exposure of indole-3-propionic acid was associated with mitochondrial dysfunction in cardiomyocytes in both mice and human hepatic and endothelial cells [160].

### 4.8. Trimethylamine N-Oxide (TMAO)

By metabolizing choline, phosphatidylcholine, L-carnitine and betaine, trimethylamine (TMA) is generated by altered gut microbiota through a range of enzymes including TMA synthase [161]. TMA is then oxidized into trimethylamine N-oxide (TMAO) in the liver by hepatic flavin monooxygenases (FMO) [162]. Changes in TMAO levels are therefore a result of changes in the composition of the gut microbiota. In chronic heart failure patients, the intestinal mucosal barrier is impaired and has an increased permeability allowing TMAO to easily enter the bloodstream and be elevated. TMAO also increases platelet reactivity through changes in stimulus-dependent calcium signaling thereby increasing atherosclerosis and thrombosis which contribute to the pathogenesis of heart failure [70]. Some studies have shown that TMAO plays an important role in modulating the gut microbiota, cholesterol metabolism and metabolic stress under cholesterol overload [163]. High concentration of TMAO activates macrophage influx of cholesterol, and affects lipid and hormonal homeostasis, eventually contributing to the development of CVDs [164]. By activating the NF-κB pathway, TMAO induces the expression of inflammatory genes in the aortic endothelial cells and vascular smooth muscle cells [161]. TMAO also upregulates the expression of vascular cell adhesion molecule-1, and promotes monocyte adherence, NF-κB and activated protein kinase C, and these effects may encourage the progression of chronic heart failure by increasing endothelial dysfunction while decreasing self-repair and activating the inflammatory response [70]. Apart from activating NF-κB, TMAO also activates NLRP3 inflammasome leading to a proinflammatory milieu that has been demonstrated in human aortic endothelial cells as well as carotid artery endothelial cells implicating TMAO to contribute to endothelial dysfunction and CVD [165,166]. Another mechanism of TMAO’s contribution to heart failure is through induction of aortic stiffness, systolic blood pressure elevation, and platelet activation resulting in a hypercoagulable state [167,168]. TMAO also worsens hypertension by directly binding to and activating protein kinase R-like endoplasmic reticulum kinase (PERK) resulting in apoptotic inflammatory responses and generation of reactive oxygen species that cause vascular injury and cardiac remodeling leading to elevated blood pressure [169,170], Figure 4. Apart from playing a role in the pathogenesis of heart failure, TMAO has also been linked to the development of several cardiovascular, metabolic and cerebrovascular conditions [171,172,173].

Choline, L-carnitine and betaine from diet are converted to trimethylamine by the gut microbiota which is then converted to trimethylamine N-oxide (TMAO) by flavin-containing monooxygenase 3 (FMO3) in the liver. TMAO activates multiple intracellular signaling pathways that promote vascular and cardiovascular pathological changes leading to heart failure. PERK, Protein kinase RNA-like endoplasmic reticulum kinase; NF-κB, nuclear factor kappa-light-chain-enhancer of activated B cells; NLRP3, NLR Family Pyrin Domain Containing 3.

## 5. Beneficial Dietary Interventions and Therapy to Modulate the Gut Microbiota in Heart Failure and Other Cardiovascular Diseases

Dietary interventions and supplementation of the gut microbiota with pre-, pro-, post-and syn-biotics and fecal transplantation is one of the available effective therapeutic approaches being used to modulate the gut microbiota for beneficial effects and to ameliorate the pathogenicity of cardiovascular and other diseases [52,174]. The Mediterranean diet and the Dietary Approaches to Stop Hypertension (DASH) diet for example have been shown to lower the risk of CVDs partly because of their modulating effect on the gut microbiota through their rich content of polyphenols, antioxidants, and mono- and polyunsaturated fatty acids which increase the levels of SCFAs [175,176,177,178]. A typical healthy Mediterranean diet would comprise mainly high amounts of fruits, vegetables, legumes, fish, nuts, cereals, grains and extra virgin oil [179].

Generally, a plant-based diet has the best beneficial effects on overall health including lowering the risk for CVDs [180]. The gut microbiota metabolizes a large quantity of the food from a plant diet to generate several metabolites that have anti-inflammatory, anti-hypertensive, antioxidant, anti-obesogenic and hypocholesterolemia effects [181]. In a study comparing 268 non-diabetic individuals stratified into strict vegetarian, lacto-ovo-vegetarian, and omnivore groups, they found that *Firmicutes* and inflammatory markers were lower while *Bacteroidetes* were higher in strict vegetarians when compared to lacto-ovo-vegetarians and omnivores [182]. In an experimental study of humanized mice harboring gut microbiota from humans, parsley and rosemary essential oils had a lowering effect on plasma INF-ɣ, TNF-α, IL-12p70 and IL-22 and modulated the gut microbiota, resulting in beneficial effects on cardiovascular and metabolic profile [183].

Extra virgin oil, a significant component of the Mediterranean diet, is rich in monounsaturated fatty acids, polyphenols and other metabolites such as hydroxytyrosol, oleuropein, tyrosol, lignans and secoirodoids [184]. Extra virgin oil is reported in many studies to lower the risk for the development of diabetes mellitus, stroke and coronary heart disease, and to improve the metabolic and inflammatory biomarker profile [179]. Some of the underlying mechanisms mediated by extra virgin oil metabolites which are beneficial in heart failure and other CVDs include: regulation of platelet aggregation and coagulation by suppressing tissue factor, coagulation factor VII, tissue plasminogen activator, plasminogen activator inhibitor-1 and fibrinogen; improving endothelial function by increasing flow mediated dilatation and NO bioavailability; improving insulin sensitivity by decreasing fasting blood sugar, glycated hemoglobin, and β-cell hyperactivity; reducing inflammation by suppressing thromboxane B2, Leukotriene B4 and C-reactive protein; and reducing oxidative stress by reducing oxidative DNA damage, F2-isoprostanes and oxidized low-density lipoprotein and regulating lipid metabolism [179]. Although the exact mechanisms by which metabolites from the Mediterranean diet promote their beneficial effects remain unknown, a brief pharmacological mechanism has been described elsewhere [179].

Resveratrol, a metabolite found in red grapes, mediates its lipid lowering effect by inhibiting the transcription factor NF-κB and activating AMP-activated protein kinase (AMPK) and sirtuin 1 [185]. Resveratrol also increases resistance of cells to oxidative stress via nuclear factor erythroid 2-related factor 2 [186]. Olive oil contains polyphenols with antioxidant effects in the heart and plays its antioxidant role via the sirtuin 1 signaling pathway [187]. Additional mechanisms for a variety of metabolites from the Mediterranean diet have been reviewed in detail elsewhere [179].

Prebiotics, which are non-digestible carbohydrates such as fiber, are beneficial in promoting healthy gut microbiota content [188]. High fiber diets have been shown to promote the growth of beneficial gut microbiota species, including by increasing bacteria that produce acetate to promote lowered blood pressure and by reducing cardiac remodeling occurring in hypertension and heart failure [189]. Additional beneficial effects of prebiotics include regulation of weight by reducing obesity, improving glucose tolerance, exerting anti-inflammatory effects, and control of ROS in several inflammatory diseases and cancers [190,191,192].

Probiotics, which are live microorganisms that promote health, have been shown to regulate obesity and reduce hyperglycemia, resulting in reduced risk for metabolic and CVDs [193,194]. Examples of beneficial microorganisms in heart failure include *Bifidobacteria*, *Lactobacillus*, *Akkermansia*, and yeasts [195]. In a randomized clinical trial involving ninety participants, Pourrajab et al. found that probiotic yogurt suppressed oxidative reactions resulting in the reduction of oxidized low-density lipoprotein cholesterol in heart failure [196]. Although the underlying mechanisms are yet to be known, these studies demonstrate the beneficial effects of dietary supplements to modulate the gut microbiota and improve heart failure.

Another effective intervention is transplantation of healthy fecal microbiota which has been beneficial to restore microbial homeostasis in irritable bowel syndrome [197], to control *Clostridium difficile* infection [198,199], and to prevent aged-related atrial fibrillation in rats by targeting and inhibiting the NLRP3-inflammasome that plays a role in promoting cardiac dysfunction [200].

## 6. Gut Microbiota Benefits on Other Organs/Systems

The gut microbiota has been shown to play an important role in various organs and systems of the body. For example, the gut–brain axis (GBA) plays a bidirectional role between the brain and the gut microbiota via signaling pathways involving endocrine, neurocrine and immune systems [201,202,203]. Modulation of the gut microbiota by a well-balanced diet therefore has beneficial effects in altering the enteric nervous system and changing the course of several neurological disorders such as Parkinson’s disease, Alzheimer’s disease, and multiple sclerosis [204,205]. In Parkinson’s disease, use of antibiotics improves behavioral symptoms suggesting that specific gut microbiota species promote this disease [206].

Gut microbiota is beneficial in many organs of the body. For example, healthy gut microbiota is important in maintaining a healthy gut barrier function in the gastrointestinal tract [207], improves kidney function by reducing indoxyl sulphate levels and improves insulin sensitivity in all cells of the body [69]. The gut microbiota also helps both to shape and enhance protection from colonizing bacteria in the integumentary system [208,209] and to promote pulmonary health [210]. Moreover, the gut microbiota plays a pivotal role in the biosynthesis of hepatic membrane phospholipids and liver regeneration [211].

## 7. Conclusions and Future Perspectives

The gut microbiota and its metabolites play a critical role in the pathogenesis of heart failure through complex signaling pathways and interactions. More investigations, especially in human studies, are required to further understand their clinical usage and potential therapeutic impact on heart failure patients. In addition to heart failure, the gut microbiota has also been implicated in various metabolic, neurological and cardiovascular diseases. Therefore, modulating the gut microbiota is beneficial not only to cardiac health but also to multiple organs and systems of the body, promoting an overall healthy milieu that ameliorates pathogenic processes.

A healthy diet, for example, is able to reduce inflammation and promote healthy gut microbiota biodiversity that promotes cardioprotective effects and also limits the progression of metabolic, cardiovascular and neurological diseases. Dietary interventions are therefore promising non-pharmacologic therapeutic approaches that patients could benefit from and should be at the core of interventional studies. Future studies need to focus on the clinical application of several therapeutic interventions that have proved to be beneficial so far in order to reduce the prevalence of cardiovascular diseases.

## Figures and Tables

**Figure 1 biomedicines-11-02313-f001:**
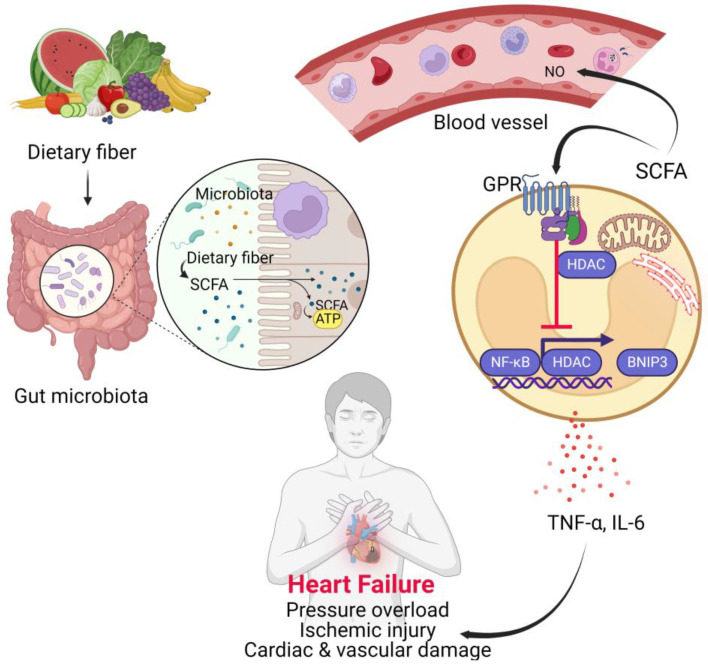
Role of short chain fatty acids in heart failure. Short chain fatty acids (SCFAs) are synthesized from fiber through gut microbiota fermentation. SCFAs provide energy to enterocytes and innate cells. On endothelial cells and innate immune cells, through signaling via the G-protein coupled receptor (GPRs), SCFAs repress nuclear factor kappa-light-chain-enhancer of activated B cells (NF-κB) in conjunction with histone deacetylases (HDACs) to inhibit Bcl-2 interacting protein 3 (BNIP3) expression and so prevent the production of pro-inflammatory cytokines that contributes to cardiac and vascular damage resulting in pressure overload and ischemic injury and heart failure. TNF-α, tumor necrosis factor alpha; NO, nitric oxide; ATP, adenosine triphosphate.

**Figure 2 biomedicines-11-02313-f002:**
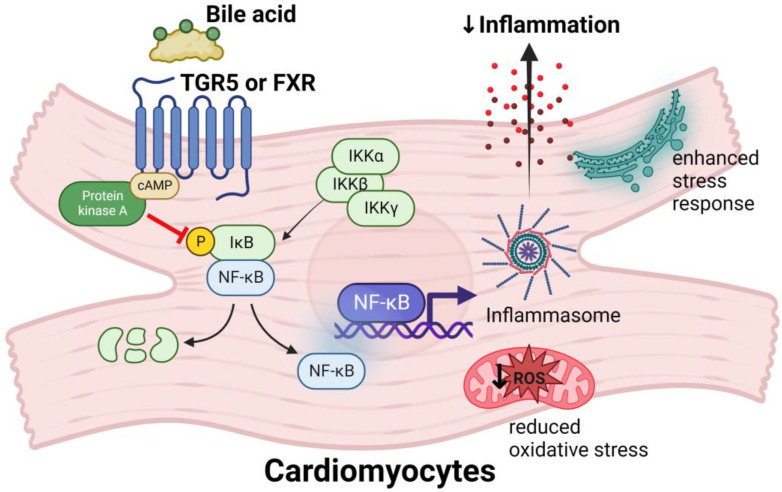
Bile acid signaling effect on FXR and TGR5 receptors on cardiomyocytes. Bile acids interact with Takeda G-protein-coupled receptor 5 (TGR5) and farnesoid X receptor (FXR) on cardiomyocytes to activate intracellular signaling pathways that promote improved cardiac function.

**Figure 3 biomedicines-11-02313-f003:**
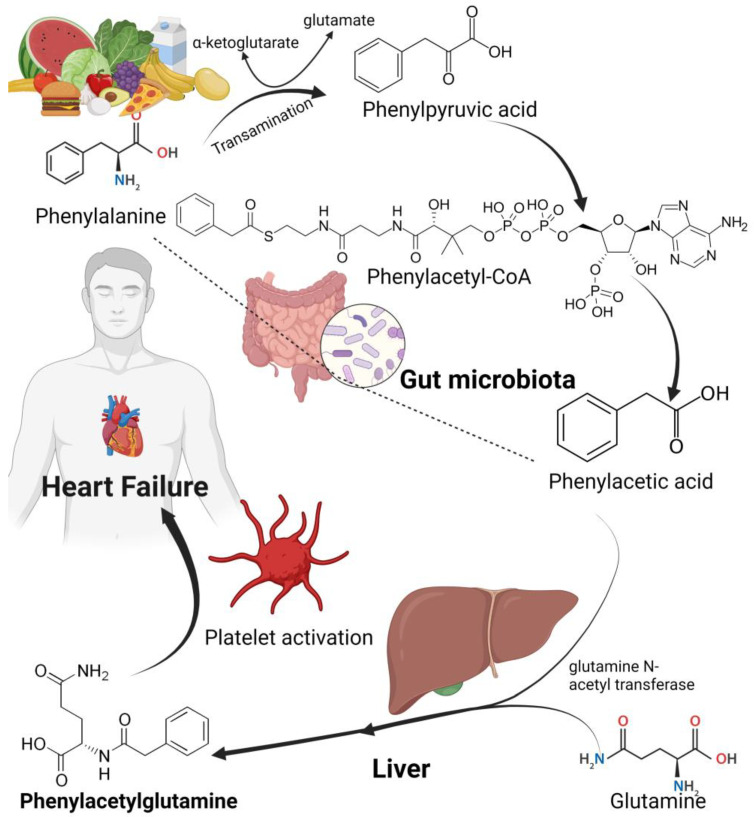
Formation of phenylacetylglutamine by the gut microbiota and liver enzymes.

**Figure 4 biomedicines-11-02313-f004:**
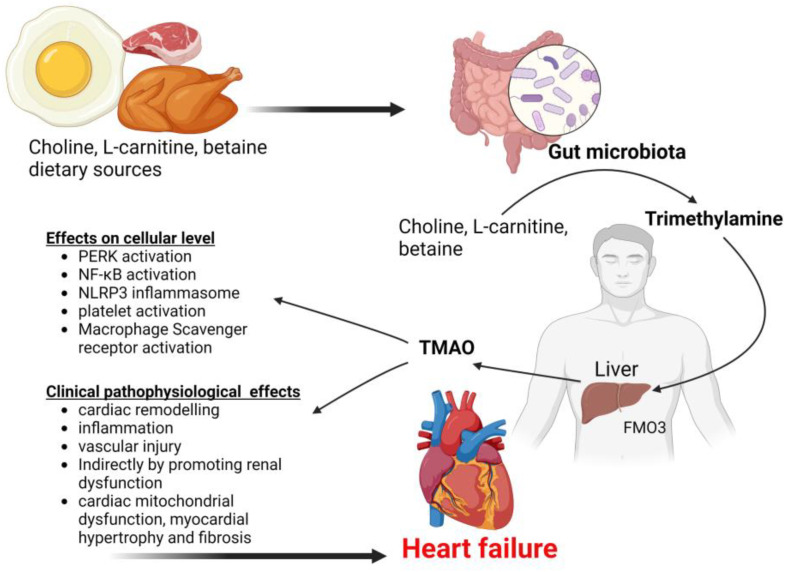
Proposed mechanisms of heart failure pathogenesis mediated by TMAO.

## Data Availability

All data presented is contained within the manuscript.

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
