# Peer review of "Recent Advances in Microbiota-Associated Metabolites in Heart Failure"

_biomedicines, 2023, doi:10.3390/biomedicines11082313_

Round 1

Reviewer 1 Report

The authors present an updated review on the effects of microbiota-derived metabolites and heart failure. In this regard, the review is up-to-date, it is well written, the structure is adequate. However, I have some comments.

I. Major comments:

1. I suggest including a short paragraph or section on the pathophysiology of heart failure and link it to the potential beneficial effects of compounds derived from the microbiota (antioxidant effects, anti-inflammatory effects, etc.)

2. It is necessary to discuss the molecular mechanisms that would allow a better understanding of the potential benefits. For example, anti-inflammatory effects through decreased activity of the transcription factor NF-kB.

3. In the figures (if possible) I suggest including mechanisms.

4. What potential application or clinical research could be projected. For example, use of pre or probiotics that modify the microbiota and can protect the heart?

5. It would be interesting to briefly mention the possible beneficial effects of the microbiota on other organs.

6. I suggest including a brief paragraph on the relationship between diet, microbiota and microbiota metabolites.

II. Minor comments:

1. Improve the writing of the objective of the manuscript.

2. Correct some writing errors.

The manuscript is well written, but some minor editorial errors need to be corrected.

Author Response

Response to reviewer’s comments

We want to thank the reviewers for the constructive feedback on this manuscript. We have made extensive revisions. The revised manuscript is improved and we hope it is suitable for publication. Below are the point-by-point responses to the reviewer comments. A Track changes version has been resubmitted.

Reviewer 1

The authors present an updated review on the effects of microbiota-derived metabolites and heart failure. In this regard, the review is up-to-date, it is well written, the structure is adequate. However, I have some comments.

Response: We thank you for the commendations

  1. Major comments:
  2. I suggest including a short paragraph or section on the pathophysiology of heart failure and link it to the potential beneficial effects of compounds derived from the microbiota (antioxidant effects, anti-inflammatory effects, etc.)

Response: Thank you for the suggestion. We have included this under the section “Beneficial effects of gut microbiota-derived metabolites in heart failure pathophysiology” and also under the section 5.0 “Beneficial dietary interventions and therapy to modulate gut microbiota in heart failure and other cardiovascular diseases”

  1. It is necessary to discuss the molecular mechanisms that would allow a better understanding of the potential benefits. For example, anti-inflammatory effects through decreased activity of the transcription factor NF-kB.

Response: We have explained and added several paragraphs in different new sections to discuss potential benefits from better understanding of molecular mechanisms including NFAT, MAPK, wnt, NF-kB etc

  1. In the figures (if possible) I suggest including mechanisms.

Response: Because some mechanisms are highlighted in the figures, we have added some text to further enhance this aspect in trying to avoid overcrowding the figures

  1. What potential application or clinical research could be projected. For example, use of pre or probiotics that modify the microbiota and can protect the heart?

Responses: This has been added under the section 5.0 “Beneficial dietary interventions and therapy to modulate gut microbiota in heart failure and other cardiovascular diseases”

  1. It would be interesting to briefly mention the possible beneficial effects of the microbiota on other organs.

Responses: we have done so under section 6.0 Gut microbiota benefits on other organs. Thank you

  1. I suggest including a brief paragraph on the relationship between diet, microbiota and microbiota metabolites.

Responses: We have included a section where this is discussed briefly 5.0 “Beneficial dietary interventions and therapy to modulate gut microbiota in heart failure and other cardiovascular diseases”

  1. Minor comments:
  2. Improve the writing of the objective of the manuscript.

Response: We have done this in the last paragraph of the introduction. Thank you for the suggestion

  1. Correct some writing errors.

Response: we have corrected all errors we could identify in the manuscript. Many thanks for your time to evaluate and highlight areas that have helped in improving our manuscript

Reviewer 2 Report

Dear authors, thank you to allow me to read this interesting and original paper focused on the central role of  microbiota and metabolites in hearth failure. The role of gut microbiota in the pathogenesys of neurodegenerative diseas like a Parkinson Disease or others is supported by many litterature data. Please link in the conclusions the role of gut microbiota in neurological and cardiovascular disease.

The conclusion are short. I suggest to add the possible future therapeutic implications and/or your personal idea and considerations.

The figures are original and well organized.

It would be appropriate to add a paragraph dedicated to diet advice, in other words which foods or food associations are recommended to reduce the activation and inflammation of the microbiota?

Finally I suggest to delete in the introduction the sentence "gut microbiota and metabolites" because is repeated too many times.

Author Response

Response to reviewer’s comments

We want to thank the reviewers for the constructive feedback on this manuscript. We have made extensive revisions. The revised manuscript is improved and we hope it is suitable for publication. Below are the point-by-point responses to the reviewer comments. A Track changes version has been resubmitted

Reviewer 2

Dear authors, thank you to allow me to read this interesting and original paper focused on the central role of microbiota and metabolites in hearth failure. The role of gut microbiota in the pathogenesis of neurodegenerative disease like a Parkinson Disease or others is supported by many literature data. Please link in the conclusions the role of gut microbiota in neurological and cardiovascular disease.

Response: Thank you for your comment. We have now mentioned the role of gut microbiota in neurological and cardiovascular disease in the conclusions

The conclusion are short. I suggest to add the possible future therapeutic implications and/or your personal idea and considerations.

Response: we have added therapeutic implications in the section on diet and have also expanded the conclusion with some recommendations and personal reflections in tandem with the topic

The figures are original and well organized.

Response: Thank you for noticing.

It would be appropriate to add a paragraph dedicated to diet advice, in other words which foods or food associations are recommended to reduce the activation and inflammation of the microbiota?

Response: We have added a new section to discuss this. See 5.0 Beneficial dietary interventions and therapy to modulate gut microbiota in heart failure and other cardiovascular diseases. Thank you

Finally, I suggest to delete in the introduction the sentence "gut microbiota and metabolites" because is repeated too many times.

Response: We have done this. Thank you

Round 2

Reviewer 1 Report

Authors answered all my comments. Therefore, the manuscript can be accepted in the present form.